

# Stable isotope analyses identify trophic niche partitioning between sympatric terrestrial vertebrates in coastal saltmarshes with differing oiling histories

Sydney Moyo[1,2], Hayat Bennadji[1], Danielle Laguaite[1], Anna A. Pérez-Umphrey[3], Allison M. Snider[3], Andrea Bonisoli-Alquati[4], Jill A. Olin[5], Philip C Stouffer[3], Sabrina S. Taylor[3], Paola C. López-Duarte[6], Brian J. Roberts[7], Linda Hooper-Bui[8] and Michael J. Polito[1]

[1] Department of Oceanography and Coastal Sciences, Louisiana State University, Baton Rouge, LA, United States of America
[2] Department of Biology, Rhodes College, Memphis, TN, United States of America
[3] School of Renewable Natural Resources, Louisiana State University and AgCenter, Baton Rouge, LA, United States of America
[4] Department of Biological Sciences, California State Polytechnic University - Pomona, Pomona, CA, United States of America
[5] Great Lakes Research Center, Michigan Technological University, Houghton, MI, United States of America
[6] Department of Biological Sciences, University of North Carolina at Charlotte, Charlotte, NC, United States of America
[7] Louisiana Universities Marine Consortium, Chauvin, LA, United States of America
[8] Department of Environmental Sciences, Louisiana State University, Baton Rouge, LA, United States of America

Corresponding author
Sydney Moyo, sydmoyo@gmail.com

## ABSTRACT

Bioindicator species are commonly used as proxies to help identify the ecological effects of oil spills and other stressors. However, the utility of taxa as bioindicators is dependent on understanding their trophic niche and life history characteristics, as these factors mediate their ecological responses. Seaside sparrows (*Ammospiza maritima)* and marsh rice rats (*Oryzomys palustris*) are two ubiquitous terrestrial vertebrates that are thought to be bioindicators of oil spills in saltmarsh ecosystems. To improve the utility of these omnivorous taxa as bioindicators, we used carbon and nitrogen stable isotope analysis to quantify their trophic niches at saltmarshes in coastal Louisiana with differing oiling histories. We found that rats generally had lower trophic positions and incorporated more aquatic prey relative to seaside sparrows. The range of resources used (i.e.,trophic niche width) varied based on oiling history. Seaside sparrows had wider trophic niches than marsh rice rats at unoiled sites, but not at oiled sites. Trophic niche widths of conspecifics were less consistent at oiled sites, although marsh rice rats at oiled sites had wider trophic niches than rats at unoiled sites. These results suggest that past oiling histories may have imparted subtle, yet differing effects on the foraging ecology of these two co-occurring species. However, the temporal lag between initial oiling and our study makes identifying the ultimate drivers of differences between oiled and unoiled sites challenging. Even so, our findings provide a baseline quantification of the trophic niches of sympatric seaside sparrows and marsh rice rats that will aid in the use of

these species as indicators of oiling and other environmental stressors in saltmarsh ecosystems.

## INTRODUCTION

Large oil spills have devastating effects on wildlife in coastal ecosystems (*Ridoux et al., 2004*; *White et al., 2012*; *Troisi, Barton & Bexton, 2016*). The *Deepwater Horizon* oil spill (DWH), which lasted for several months, was the largest (>700,000 m$^3$ of crude oil; *McNutt et al., 2012*) marine oil spill in the history of the United States of America (*Crone & Tolstoy, 2010*). This spill resulted in high mortalities of birds, sea turtles, and marine mammals (*Antonio, Mendes & Thomaz, 2011*). Additionally, DWH affected ~2,113 km of the Louisiana coastline, including ecologically and economically important coastal marshes (*Nixon et al., 2016*); coastal wetlands accounted for at least 1,105 km (52%) of oiled shorelines, 95% of which were in Louisiana (*Nixon et al., 2016*). Researchers have documented that oil penetrated into the marsh with zones of oiled plant canopies extending an average of 11 m from the shoreline, with a maximum penetration distance of 21m (*Kokaly et al., 2013*; *Michel et al., 2013*; *Zengel et al., 2015*). In 2011, oil in some locations was found to have penetrated more than 100 m into the marsh (*Turner et al., 2014b*). Studies documenting the effects of stressors such as oil spills, have conventionally focussed on the acute effect rather than the chronic effects on vertebrates (*Helm et al., 2015*). As species can differ in their response to common stressors, there is a need for studies that investigate the long-term (indirect) effects of stressors on co-occurring species.

Terrestrial vertebrates are commonly used as proxies of contaminant effects in coastal wetland ecosystems (*Frederick, Spalding & Dusek, 2002*; *Rabalais & Turner, 2016*), because shoreline contamination increases the probability of terrestrial animals being exposed to such contaminants (*McCann et al., 2017*) particularly because of the depth of penetration of the oil (see *Turner et al., 2014b*). Seaside sparrows (*Ammospiza maritima*) and marsh rice rats (*Oryzomys palustris*) may be useful bioindicators in saltmarshes along the northern Gulf of Mexico (reviewed by *Bergeon Burns et al., 2014*). Both are ubiquitous in these systems, and are generalist omnivores that exhibit high site fidelity (*Kern & Shriver, 2014*). Owing to their high fidelity, oiling can have differential effects on seaside sparrows (hereafter referred to as 'sparrows') and marsh rice rats (hereafter referred to as 'rats') . For example, the DWH oil spill caused sparrow mortality (*USFWS, 2011*) and altered expression of genes involved in the liver's metabolism of xenobiotics (*Perez-Umphrey et al., 2018*), as well as liver regeneration and homeostasis (*Bonisoli-Alquati et al., 2020*). Similarly, the DWH oil spilled resulted in sparrows incorporating carbon from oil into gut contents and newly grown feathers (*Bonisoli-Alquati et al., 2016*). The DWH oil spill led to decreased insect and arthropod abundances and altered community compositions at oiled sites compared to unoiled sites (*Zengel et al., 2015*; *Husseneder, Donaldson & Foil, 2016*; *Bam et al., 2018*)

possibly leading to increased competition among terrestrial vertebrates for prey at oiled sites (*sensu Bergeon Burns et al., 2014*). Despite their potential as bioindicators, limited information exists on the trophic niches and trophic interactions of sparrows and rats (although see *Olin et al., 2017*; *Johnson, Olin & Polito, 2019*).

Species-specific feeding ecology and life history need to be well characterized for bioindicator approaches to be effective (*Greenberg et al., 2006*). Food web approaches describing the trophic links between organisms can provide information about species-specific behaviors and potential effects of contaminants on co-occurring species in an ecosystem (*O'Gorman, Fitch & Crowe, 2012*; *Kovalenko, 2019*). Trophic niches often differ among co-occurring species (e.g., *Polito et al., 2016*; *Dionne, Dufresne & Nozais, 2017*) and are influenced by factors such as morphology (*Maia-Carneiro, Motta-Tavares & Rocha, 2017*), physiology (*Franco-Trecu, Aurioles-Gamboa & Inchausti, 2014*), foraging strategy (*Blanchet-Aurigny et al., 2015*), food availability (*Akasaka, Nakano & Nakamura, 2009*) and competition (*Mwijage, Shilla & Machiwa, 2018*). These factors can lead to interspecific niche partitioning that supports co-existence of species over time (*Hutchinson, 1959*; *May & Arthur, 1972*; *Olin et al., 2020*). For example, rats and sparrows use different foraging strategies, with sparrows known to aerially hawk and glean off substrate (e.g., vegetation, marsh sediment, etc.) for their food (*Post & Greenlaw, 2006*). Conversely, rats feed on terrestrial and aquatic resources available on the marsh platform (*Kruchek, 2004*). Sparrows are strictly diurnal while rats are nocturnal (*Post, 1981*). These different foraging strategies may also lead to different resource use and thus niche partitioning between sparrows and rats. However, no prior study has concurrently examined the trophic niches of these two species in sympatry. Knowledge of resource use and overlap between sympatric species is important not only from an ecological context but also has a significant bearing on assessing the potential effect of future environmental disturbances (*sensu Fenton, 2003*; *Hunter et al., 2015*). Therefore, an improved understanding of the trophic ecology of sparrows and rats is warranted both to examine the potential for niche partitioning and to improve the use of these species as bioindicators of past and future environmental perturbations (*Bergeon Burns et al., 2014*; *Bonisoli-Alquati et al., 2016*; *Bonisoli-Alquati et al., 2020*; *Olin et al., 2017*; *Pérez-Umphrey et al., 2018*).

Stable isotope analysis offers an effective tool for quantifying the trophic ecology of a wide range of consumers. This approach has been used in past studies to examine the trophic niche of coastal consumers (*Blanchet-Aurigny et al., 2015*; *Zhang et al., 2019*), the dietary overlap of co-occurring vertebrate species (*Bootsma et al., 1996*; *Larson, Twardochleb & Olden, 2017*), and the relative importance of basal carbon sources in consumer food webs (*Garcia et al., 2017*; *Grieve & Lau, 2018*; *David et al., 2019*). Nitrogen stable isotope values ($\delta^{15}$N) are commonly used to infer the trophic position of consumers, while carbon stable isotope values ($\delta^{13}$C) can be used to quantify the use of terrestrial vs. aquatic resources (*Crawford, Mcdonald & Bearhop, 2008*; *Inger & Bearhop, 2008*). Moreover, in combination, $\delta^{13}$C and $\delta^{15}$N values act to define a consumer's isotopic niche, a proxy of a consumer's trophic niche, which can be compared across species, time, and space (*Bearhop et al., 2004*; *Newsome et al., 2007*).

We used stable isotope analysis to quantify and compare the trophic position, relative importance of terrestrial vs. aquatic resources, and trophic niche width and overlap of co-occurring sparrows and rats at saltmarshes in Barataria Bay, Louisiana. Differing morphologies, physiologies and behaviours often lead co-occurring species to use different resources and occupy separate and unique trophic niches (i.e., niche partitioning; *Fink et al., 2012*; *Karlson, Gorokhova & Elmgren, 2015*; *Rocha, Bini & Siqueira, 2018*). As such, we hypothesized that: (1) sparrows and rats occupy similar trophic positions because they are both omnivores; (2) rats incorporate more aquatic prey than sparrows owing to their ground foraging strategies and swimming; (3) sparrows have wider trophic niches compared to rats because sparrows are diurnal and feed both aerially and on the ground, and (4) sparrows and rats exhibit asymmetric niche overlap as a function of their hypothesized similar trophic positions yet differing resource use and trophic niche widths. Specifically, trophic niches of rats will exhibit greater overlap with the trophic niches of sparrows than *vice versa*. Here, niche overlap is the probability that an individual from one species is found within the total trophic niche of the other species. We considered these four hypotheses across sites with differing oiling histories following the 2010 DWH oil spill, to examine the chronic effects and ecological legacy of the disaster on these species' trophic interactions and species-specific trophic niches.

## MATERIAL AND METHODS

### Study species and life histories

Seaside sparrows are the most common small passerines (∼15–25 g) found in Louisiana coastal marshes (*Grenier & Greenberg, 2006*). Sparrows rely on seeds, insects, and other marsh invertebrates (*Post & Greenlaw, 2006*). Both sparrows and rats are year-round residents with small home ranges (*Cooney, Schauber & Hellgren, 2015*), especially during the breeding season, making them highly susceptible to a wide range of environmental perturbations to coastal marsh ecosystems (e.g., flooding, climate change, sea rise, pollution; (*Bergeon Burns et al., 2014*; *Kern & Shriver, 2014*). Of the 1,185 sparrows captured from our study sites between 2011–2017, 79 sparrows were captured in more than one sampling year. While a majority of recaptures occurred one or two years apart ($n = 43, 24$ respectively), six birds were captured across a time span of three years while six were captured across a time span of four years. Notably, these sparrows were adults when initially captured, so these birds were likely older than three or four years of age. The oldest wild sparrow recorded was at least nine years old (*Sykes, 1980*); while this lifespan is likely uncommon, it may be possible that a small number of birds sampled in this study could have been alive during the DWH spill. No nestlings were recaptured as adults during our study.

Marsh rice rats are mid-sized (∼40–80 g), semi-aquatic cricetid rodents common in Louisiana coastal marshes (*Wolfe, 1982*). Rats are omnivorous and known to consume a range of prey with plants, crabs, insects and molluscs making significant contributions to their diet (*Sharp, 1967*; *Kruchek, 2004*). Rat populations are characterized by turnover rates, with rats rarely living longer than one year, although one individual was documented to live up to two years (*Wolfe, 1985*). At our study sites, only three rice rats were recaptured

over two consecutive field seasons (A. Perez-Umphrey, unpublished data), one in 2014 then in 2015, two in 2016 then in 2017 (although, note, 2014 samples were not included in this present study). Because only subadult and adult rats were bled, and all of these individuals were juveniles in their first year of capture; only a single sample exists for each of these three individuals from when they were recaptured. No individuals sampled would have been present in 2010.

## Ethics statement

All animals (birds, rats, and invertebrates) and plants were sampled on private or public land after obtaining prior permission from owners or managers, and the necessary permits. The capture and handling of all animals was approved by the United States Fish and Wildlife Service (collecting permits MB095918-0 and MB095918-1; USFWS Federal Bird Banding permit 22648; and, Louisiana Department of Wildlife and Fisheries scientific collecting permits LNHP-15-039, LNHP-16-056, LNHP-17-064, LNHP-15-033, LNHP-16-048, LNHP-17-039). All sampling protocols adhered to statutes of the Institutional Animal Care and Use Committee of the LSU AgCenter (IACUC permits: A2013-09, A2015-04, A2016-06).

## Study area

Samples were collected from seven saltmarsh sites in the Barataria Bay region of coastal Louisiana, USA between April and June of 2015, 2016, and 2017 (Fig. 1). The saltmarsh in this area is either dominated by smooth cordgrass, *Spartina alterniflora* or less commonly co-dominated with *Juncus roemerianus*. This region of Barataria Bay was selected as the focus of this study because it received some of the heaviest oiling following the 2010 DWH oil spill (*Michel et al., 2013*; *Turner et al., 2014a*). We selected seven marsh sites based on their oiling histories as determined by a combination of systematic methods [Shoreline Clean-up and Assessment Technique (SCAT); (*Michel et al., 2013*)] and chemical analyses (*Turner et al., 2014a*) following the 2010 spill. These seven sites laid the basis for our experimental design in 2015, 2016, and 2017. We categorized sites as either 'oiled' ($n = 4$) or 'unoiled' ($n = 3$) based on the SCAT maps' documented categories of oiling (categories: heavy, moderate, light, very light, trace oiling and no oiling; *Turner et al., 2014a*) from the marsh edge. Our sites were in Bay Batiste (O2, O3, O4- heavily oiled), Bay Sanbois (U1,U2- unoiled), and Bay Jimmy (O1- heavily oiled, U3- unoiled). Sampling plots within sites were 2.5 ha (500 m along the marsh edge and 50 m inland) and sites were a minimum of 1 km apart (Fig. 1). Site classification was confirmed through sediment analyses (*Turner et al., 2014a*; *Turner et al., 2014b*)—oiled marsh sites had initial polycyclic aromatic hydrocarbon (PAH) mean $\pm$ standard deviation concentrations of 19,524 $\pm$ 2,158 $\mu g\,kg^{-1}$ (*Turner et al., 2014a*) and unoiled sites had initial (PAH) mean $\pm$ standard deviation concentrations of 164 $\pm$ 21 $\mu g\,kg^{-1}$ (*Ashton-Meyer, 2017*). Although our study started five years after initial oiling when oiling effects may have attenuated, we explicitly explored these comparisons to test for chronic and/or indirect effects because oil can persist in marsh habitats over long periods of time (*Turner et al., 2019*).

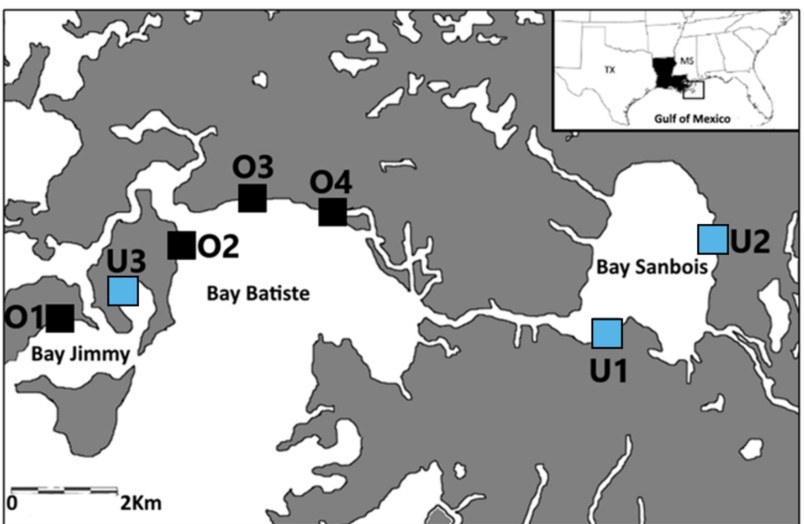

**Figure 1 Sampling locations.** Sampling locations (previously oiled versus non-oiled) in the Port Sulphur area of Barataria Bay, Louisiana.

## Sample collections

Rats were caught in baited Sherman live traps set overnight. Sparrows were caught in mist nets (34 mm mesh, 12 m length). All rats and sparrows captured were weighed, sexed, and aged. Aging of sparrows and rats ensured that only adults and sub-adults were sampled (i.e., juveniles were excluded from all subsequent analyses). Blood samples were drawn from sparrows and rats via venipuncture of the brachial vein and retro-orbital sampling, respectively. All blood samples were transferred to glass vials and stored frozen until further analysis.

Suspended particulate organic matter (POM) was collected by passing ~250mL of water collected off the marsh edge through a 105-micron screen to remove large zooplankton and inorganic particulates and onto combusted glass fibre filters (Whatman GF/F). We confirmed that we retained most of the phytoplankton community with this approach by analysing chlorophyll-*a* concentration on screened and unscreened samples. Chlorophyll-*a* concentrations on screened samples were >85% of total chlorophyll-*a* on unscreened samples (and over 90% in most cases; B. Roberts, unpublished data). Gulf ribbed mussels (*Geukensia granossissima*) were collected by hand from the marsh surface. Samples of the dominant marsh vegetation (i.e., *Spartina alterniflora*) were collected along with herbivorous terrestrial Hemiptera (*Prokelisia* spp and *Ischnodemus* spp) by hand and using sweep nets. These primary producers and primary consumer samples provided isotopic proxies of the dominant aquatic and terrestrial marsh energy pathways, respectively. Muscle tissue was dissected from ribbed mussels, whole bodies were retained for insects, and all samples were frozen in the field prior to analysis.

## Stable isotope analysis

Blood, muscle, insect, and plant samples were freeze-dried for 24 to 48 h, homogenised into a powder, and weighed into tin capsules for stable isotope analysis. Freeze-dried POM samples were removed from filters and placed into tin capsules for stable isotope analysis. Samples were flash combusted using a Costech ECS4010 Elemental Analyzer and analysed for $\delta^{13}C$ and $\delta^{15}N$ values via a coupled to a Thermo Scientific XP or Delta V Isotope Ratio Mass Spectrometer. Stable isotope ratios are expressed in $\delta$ notation in per mil units (‰), according to the following equation:

$$\delta X = [(R_{sample}/R_{standard}) - 1] \times 1000$$

Where X is $^{13}C$ or $^{15}N$ and R is the corresponding ratio $^{13}C/^{12}C$ or $^{15}N/^{14}N$. The $R_{standard}$ values are referenced to the Vienna PeeDee Belemnite (VPDB) for $\delta^{13}C$ and atmospheric $N_2$ for $\delta^{15}N$.

Raw $\delta$ values were normalized on a two-point scale using glutamic acid reference materials with low and high values [i.e., USGS40 ($\delta^{13}C = -26.4‰$, $\delta^{15}N = -4.5‰$) and USGS41 ($\delta^{13}C = 37.6‰$, $\delta^{15}N = 47.6‰$)]. The analytical precision, based on standard deviations of repeated reference materials were 0.1‰ and 0.2‰ for $\delta^{13}C$ and $\delta^{15}N$, respectively.

## Data analyses

We performed a three-way analysis of similarities (ANOSIM), which permits unbalanced replication between treatments (Clarke & Gorley, 2006), to describe differences in the isotope values of basal end members and primary consumers. Taxa, year, and treatment (site) were classified as factors in all analyses and resemblance matrices were based on Euclidean distances (9999 permutations). ANOSIM, a non-metric multivariate statistical method, has no underlying assumptions about the statistical distribution of the data (e.g., normality, variance equality) and creates an overall test statistic (R) that indicates if differences among taxa, sites and years exist (Clarke & Gorley, 2006). ANOSIM results in an R-value and a p-value. The R-value is scaled from −1 to +1. R-values = 0 indicate random grouping, $R \geq 0.3$ shows that groups are slightly different but overlapping, and $R \geq 0.5$ indicates well separated groups (Clarke & Gorley, 2006).

We calculated trophic position (TP) of sparrows and rats using Bayesian models using mode for TP estimates in the tRophicPosition package (Quezada-Romegialli et al., 2018) in the statistical software R (version 3.5.0; R Core Team, 2019). This approach couples Markov Chain Monte Carlo simulations with stable isotope data to estimate TP. Because TPs calculated from Bayesian models display higher uncertainty with smaller sample size (Quezada-Romegialli et al., 2018), and low samples sizes were present in some species, site, and year combinations, we pooled data across all oiled and unoiled sites, respectively. In all TP calculations, we set the model parameters as follows: baseline = 'twobaselinesfull', iterations = 100000, number of chains (n. chains) = 5 and 'burn-in' (number of initial iterations discarded) = 20000. We used the stable isotope values ($\delta^{13}C$; $\delta^{15}N$) of ribbed mussels and herbivorous insects as representative of aquatic and terrestrial food web baselines, respectively. To validate our assumption that primary consumers were representative of aquatic and terrestrial resources, we visually compared them to the

stable isotope values of POM and marsh vegetation (i.e., *Spartina alterniflora;* Fig. 2). We then used tRophicPosition to quantify the relative importance of terrestrial and aquatic food resources ($\alpha$) to rats and sparrows, with larger $\alpha$ values implying greater use of terrestrial relative to aquatic food resources. We calculated TP and $\alpha$ separately for each species, site, and year combination and all models incorporated an assumed mean trophic fractionation ($\Delta^{13}C = 0.39 \pm 1.30$; $\Delta^{15}N = 3.4 \pm 0.98$) per trophic transfer (*Post, 2002*). TP and $\alpha$ metrics are presented as modal values. Associated 95% credibility intervals (CI) and Bayesian posterior probabilities (PP > 0.95) from pairwise comparisons of posterior distributions of TP and alpha ($\alpha$), which were used to indicate statistically significant differences across species, site, and year combinations. Pairwise comparisons among years between and within species were expressed as Bayesian posterior probabilities (PP > 0.95) to test for significant differences among groups. We pooled data across all oiled and unoiled sites for sparrows and rats. We used this approach for three reasons: first, preliminary analyses revealed that there were similar trends between pooled and site-specific analyses (Table S1). Second, our analyses are sensitive to sample sizes. As such, it was necessary to combine samples to avoid bias (*Jackson et al., 2011*; *Quezada-Romegialli et al., 2018*). Third, pooling samples into 'oiled' and unoiled' categories would allow us to make direct comparisons of overall differences between previously oiled and unoiled sites.

We quantified the trophic niche width of sparrows and rats using the Stable Isotope Bayesian Ellipses in R (SIBER) package version 2.1.4 (*Jackson et al., 2011*). We visualized trophic niches using the standard ellipse area corrected for sample size ($SEA_c$) represented by the bivariate standard deviation of $\delta^{13}C$ and $\delta^{15}N$ values, encompassing approximately 40% of the data (*Jackson et al., 2011*). Moreover, we compared the Bayesian approximation of standard ellipse area ($SEA_b$) between groups as a quantitative measure of isotopic trophic niche width. Because $SEA_b$ display higher uncertainty with smaller sample size (*Jackson et al., 2011*), and low samples sizes were present in some species, site, and year combinations, we pooled data across oiled and unoiled sites prior to analyses similar to our analysis of TP. Values for $SEA_b$ are reported as modal values with 95% CI and Bayesian posterior probabilities (PP>0.95) were used to test for significant differences among groups.

To quantify overlap in resource use (i.e., trophic niche overlap) among species and between sites, we quantified the proportional overlap between Bayesian-estimated standard ellipses ($SEA_b$) of each group using nicheROVER package 1.0 (*Swanson et al., 2015*). As in analysis of $SEA_b$ area, we pooled data into oiled and unoiled sites prior to analyses. Trophic niche overlap is defined as the probability that an individual from one species is found within the total trophic niche of the other species, with values ranging from 0% (no overlap) to 100% (complete overlap). We considered trophic niche overlap to be asymmetrical when 95% CI did not overlap among reciprocal comparisons of CI, such that individuals in one group are significantly more likely to be encompassed in the ellipses of another group than *vice versa*.

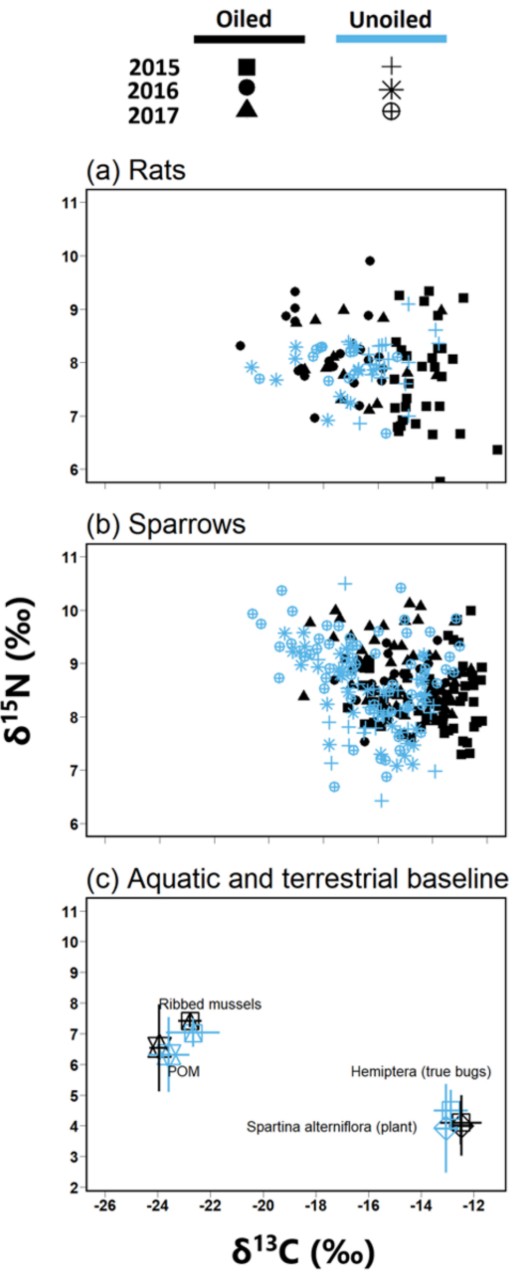

**Figure 2 Biplot of stable isotope values of fauna and flora.** Biplots (mean $\pm$ SD) of isotope values ($\delta^{13}C$ and $\delta^{15}N$) of (A) March rice rats (*O. palustris*) (B) seaside sparrows (*A. maritima*) (C) primary consumers isotopic values relative to primary producer end members collected across all three years.

# RESULTS

## Summary of basal endmembers and primary consumers

Biplot of $\delta^{13}$C and $\delta^{15}$N and ANOSIM revealed clear separations between POM and *Spartina alterniflora* (Fig. 2; Global $R = 1.0$, $p = 0.001$) with no significant differences in stable isotope values by treatment [oiled sites versus unoiled sites; ANOSIM, Global

$R = 0.1$, $p = 0.23$] or year (ANOSIM $= 0.1$, $p = 0.13$). $\delta^{15}N$ values of POM (range $= 4.3$ to $8.6‰$) were higher than *Spartina alterniflora* (range $= 1.7$ to $5.8‰$). Conversely, $\delta^{13}C$ values in POM (range $=-24.4$ to $-22.2‰$) were lower than *Spartina alterniflora* (range $=-13.6$ to $-11.8‰$).

Biplot of $\delta^{13}C$ and $\delta^{15}N$ and ANOSIM also revealed clear separations between hemipterans and ribbed mussel (Fig. 2; ANOSIM, Global $R = 0.9$, $p = 0.001$). Ribbed mussels (aquatic baseline) were similar across all sites (ANOSIM, Global $R = 0.3$, $p = 0.02$) and year (ANOSIM, Global $R = 0.2$, $p = 0.4$). Similarly, hemipterans (terrestrial baseline) did not vary by site (ANOSIM, Global $R = 0.0$, $p = 0.2$) or year (ANOSIM, Global $R = 0.0$, $p = 0.5$). Stable isotope values of ribbed mussels (aquatic baseline; $\delta^{13}C$ range $= -23.7$ to $-19.2‰$; $\delta^{15}N$ range $= 5.2$ to $7.8‰$) and hemipterans (terrestrial baseline; $\delta^{13}C$ range $= -14.0$ to $-11.2$; $\delta^{15}N$ range $= 3.3$ to $6.5$) plotted closely to POM and *Spartina alterniflora*, respectively.

### Trophic position [Hypothesis 1]
Trophic position (TP) for sparrows and rats ranged between 2.5 to 3.1 (modal values; Table 1). TP was higher in sparrows than rats at both oiled and unoiled sites in 2016 and 2017, and at oiled sites in 2015 (Table 1; Fig. 3). Trophic positions of sparrows did not differ between oiled and unoiled sites in any year examined (Table 1; Table S2; Fig. 2). Similarly, TP of rats did not differ between oiled and unoiled sites in any year examined (Table 1; Table S2; Fig. 3).

### Resource use [Hypothesis 2]
Sparrows consistently relied on terrestrial resources to a greater degree than aquatic resources (i.e., mode $\alpha > 50\%$; Fig. 3; Table 1) in all three years sampled. Rats relied more on terrestrial resources than aquatic resources in 2015. Conversely, in 2016 and 2017 rats assimilated more aquatic prey than terrestrial prey. (Fig. 3; Table 1). Considering resource use between rats and sparrows, the use of terrestrial prey resources relative to aquatic prey resources ($\alpha$) did not differ between sparrows and rats in 2015; however, in 2016 and 2017 rats incorporated more aquatic prey resources (i.e., lower $\alpha$) relative to sparrows irrespective of the oiling history of sites (Fig. 3; Table 1; Table S3).

Within species (pairwise comparison of posterior probabilities of $\alpha$; Table 1), significant differences in $\alpha$ were observed in some comparisons between oiled and unoiled sites in both sparrows and rats (Table 1; Table S3). Specifically, modal $\alpha$ values of sparrows were higher at oiled sites than unoiled sites in 2015 and 2017 but were similar in 2016 (Table 1). Modal $\alpha$ values of rats at oiled sites were greater than those at unoiled sites in 2015 but were similar in 2016 and 2017 (Table 1).

### Trophic niche width [Hypothesis 3]
Sparrows had larger isotopic niche widths (SEA$_b$) than rats at unoiled sites in all three years, and at oiled sites only in 2015 (Table 1; Fig. 3). Sparrows had smaller trophic niche widths (SEA$_b$) at oiled sites relative to unoiled sites in 2016 and 2017 (Table 1; Fig. 3). In contrast, rats at oiled sites had consistently larger trophic niche widths (SEA$_b$) relative to rats at unoiled sites in all three years examined (Table 1; Fig. 3; Table S4).

**Table 1 Stable isotope values, trophic position, posterior alpha (α) and isotopic foraging niche width (SEAb) of seaside sparrow and marsh rice rat.** Sample size (n), carbon ($\delta^{13}$C) and nitrogen ($\delta^{15}$N) stable isotope values, trophic position (TP), posterior alpha (α) and isotopic trophic niche width (SEA$_b$) of seaside sparrows (*A. maritima*) and marsh rice rats *(O. palustris)* from oiled and unoiled sites, calculated by multivariate ellipse-based metrics and Bayesian estimates (SIBER R package; *Jackson et al., 2011*), across three study periods (2015, 2016, 2017) from Barataria Bay, Louisiana. Different letters indicate significant differences in each year (PP > 0.95) based on the probability associated with paired comparisons for species at each treatment (oiled versus unoiled sites).

| Year, Species | Group | n | Mean ± SD | | Modal (95% CI) | | |
| | | | $\delta^{13}$C | $\delta^{15}$N | TP | α | SEA$_b$ |
|---|---|---|---|---|---|---|---|
| **2015** | | | | | | | |
| Rats | Oiled | 33 | −14.2 ± 0.7 | 7.6 ± 0.7 | 2.8 (2.7–2.9)[a] | 0.7 (0.7–0.8)[a] | 1.6 (1.4–2.4)[a] |
| | Unoiled | 23 | −15.7 ± 0.7 | 7.9 ± 0.3 | 2.8 (2.7–2.9)[ab] | 0.6 (0.6–0.7)[b] | 0.7 (0.6–1.1)[b] |
| Sparrows | Oiled | 60 | −14.1 ± 1.5 | 8.3 ± 0.6 | 3.0 (2.9–3.2)[b] | 0.7 (0.7–0.8)[a] | 2.7 (2.5–3.5)[c] |
| | Unoiled | 48 | −15.9 ± 1.2 | 8.1 ± 0.7 | 2.9 (2.8–3.1)[ab] | 0.7 (0.6–0.7)[b] | 2.5 (2.2–3.7)[c] |
| **2016** | | | | | | | |
| Rats | Oiled | 21 | −17.7 ± 1.4 | 8.1 ± 0.7 | 2.7 (2.5–2.8)[a] | 0.4 (0.4–0.5)[a] | 3.1 (2.7–4.9)[a] |
| | Unoiled | 15 | −17.9 ± 1.5 | 7.7 ± 0.3 | 2.5 (2.4–2.7)[a] | 0.4 (0.3–0.5)[a] | 1.4 (1.2–2.6)[b] |
| Sparrows | Oiled | 33 | −15.4 ± 1.1 | 8.6 ± 0.4 | 3.1 (3.0–3.2)[b] | 0.6 (0.6–0.7)[b] | 1.4 (1.3–2.3)[a] |
| | Unoiled | 47 | −16.6 ± 1.6 | 8.4 ± 0.7 | 2.9 (2.8–3.1)[b] | 0.6 (0.5–0.7)[b] | 3.1 (2.7–4.5)[c] |
| **2017** | | | | | | | |
| Rats | Oiled | 15 | −16.6 ± 1.4 | 8.0 ± 0.6 | 2.7 (2.6–2.9)[a] | 0.5 (0.5–0.6)[a] | 2.6 (2.2–4.6)[a] |
| | Unoiled | 12 | −17.4 ± 1.1 | 8.0 ± 0.3 | 2.7 (2.6–2.8)[a] | 0.5 (0.4–0.5)[a] | 1.0 (0.8–1.9)[b] |
| Sparrows | Oiled | 67 | −14.9 ± 1.5 | 8.8 ± 0.6 | 3.1 (3.0–3.2)[b] | 0.7 (0.6–0.7)[b] | 2.7 (2.4–3.5)[a] |
| | Unoiled | 75 | −16.3 ± 1.9 | 8.8 ± 0.8 | 3.0 (2.9–3.1)[b] | 0.6 (0.5–0.6)[c] | 4.7 (4.4–6.2)[c] |

## Trophic niche overlap [Hypothesis 4]

Trophic niche overlap between sparrows and rats was higher in 2015 than 2016 and 2017 at both oiled and unoiled sites (Fig. 4), with 2016 and 2017 showing minimal overlap between rats and sparrows at oiled and unoiled sites. When comparing niche overlap within species (rats vs rats, sparrows vs sparrow), trophic niche overlap within species was higher in 2016 and 2017 relative to 2015 for both sparrows and rats (Fig. 4).

The trophic niches of sparrows and rats exhibited a generally asymmetric pattern of niche overlap at both oiled and unoiled sites (Table 2). Specifically, rats exhibited larger trophic niche overlap relative to sparrows than *vice versa*, in all three years examined at unoiled sites (Table 2). However, at oiled sites the direction and consistency of asymmetric trophic niche overlap differed across years. For example, at oiled sites in 2015, rats exhibited larger niche overlap (95% CI [71–93]%) relative to sparrows, than *vice versa* (95% CI [49–70]%; Table 2). In contrast, at oiled sites in 2017 sparrows exhibited larger niche overlap (95% CI [39–80]%) relative to rats, than *vice versa* (95% CI [34–66]%; Table 2). Trophic niche overlap between species was symmetrical at oiled sites in 2015 (Table 2).

## DISCUSSION

In coastal marsh ecosystems, the feeding ecology of terrestrial omnivores are often flexible and dependent on environmental conditions (*Ruiter, Wolters & Moore, 2005*) similar

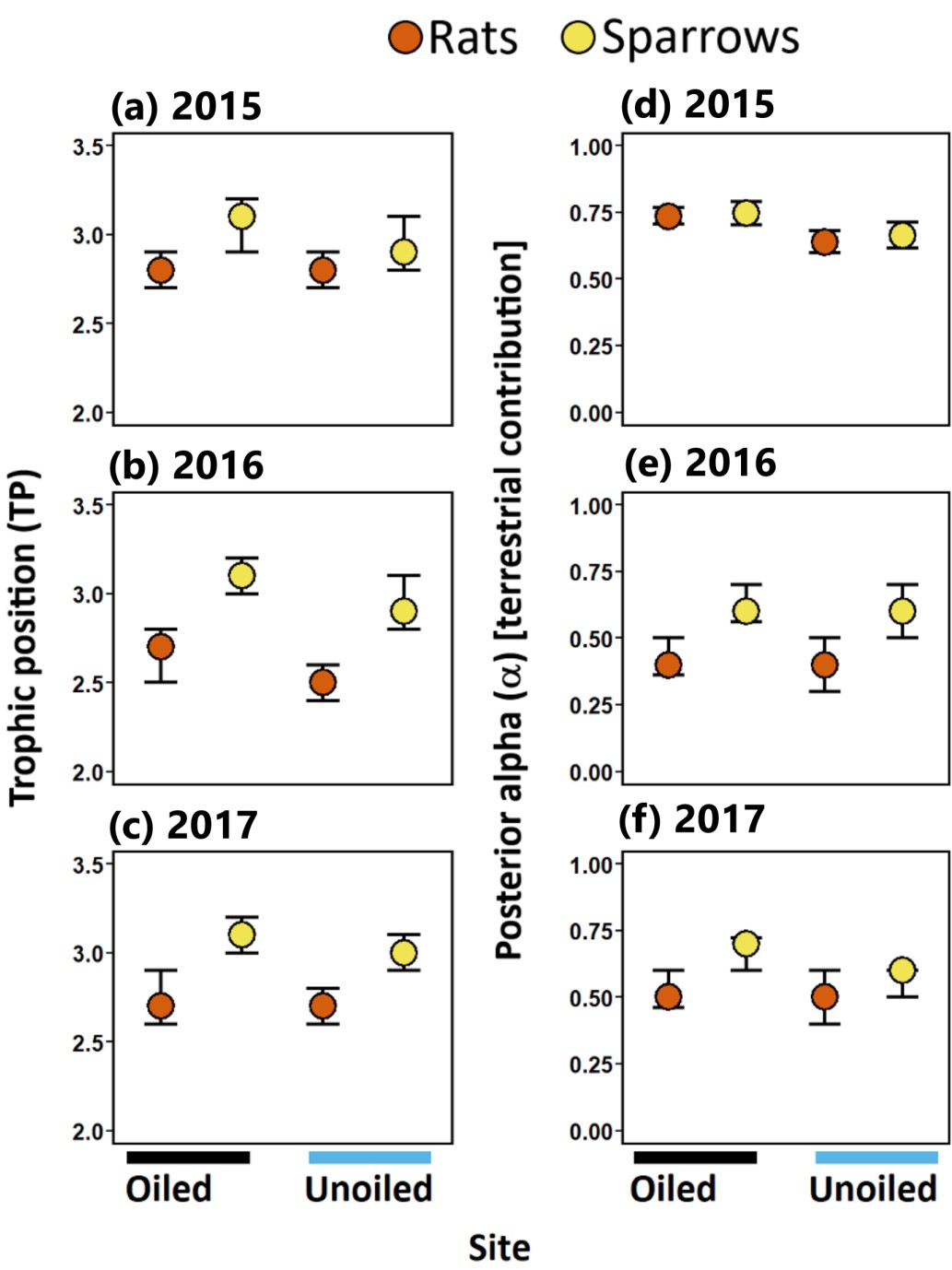

**Figure 3  Trophic positions and posterior alpha of seaside sparrow and marsh rice rat.** Modal (95% CI) trophic position (TP) [*left panel* ] and posterior alpha [estimate of importance of terrestrial food] [*right panel* ] of sympatric seaside sparrow (*A. maritima*) and marsh rice rat (*O. palustris*) assuming terrestrial and aquatic baselines.

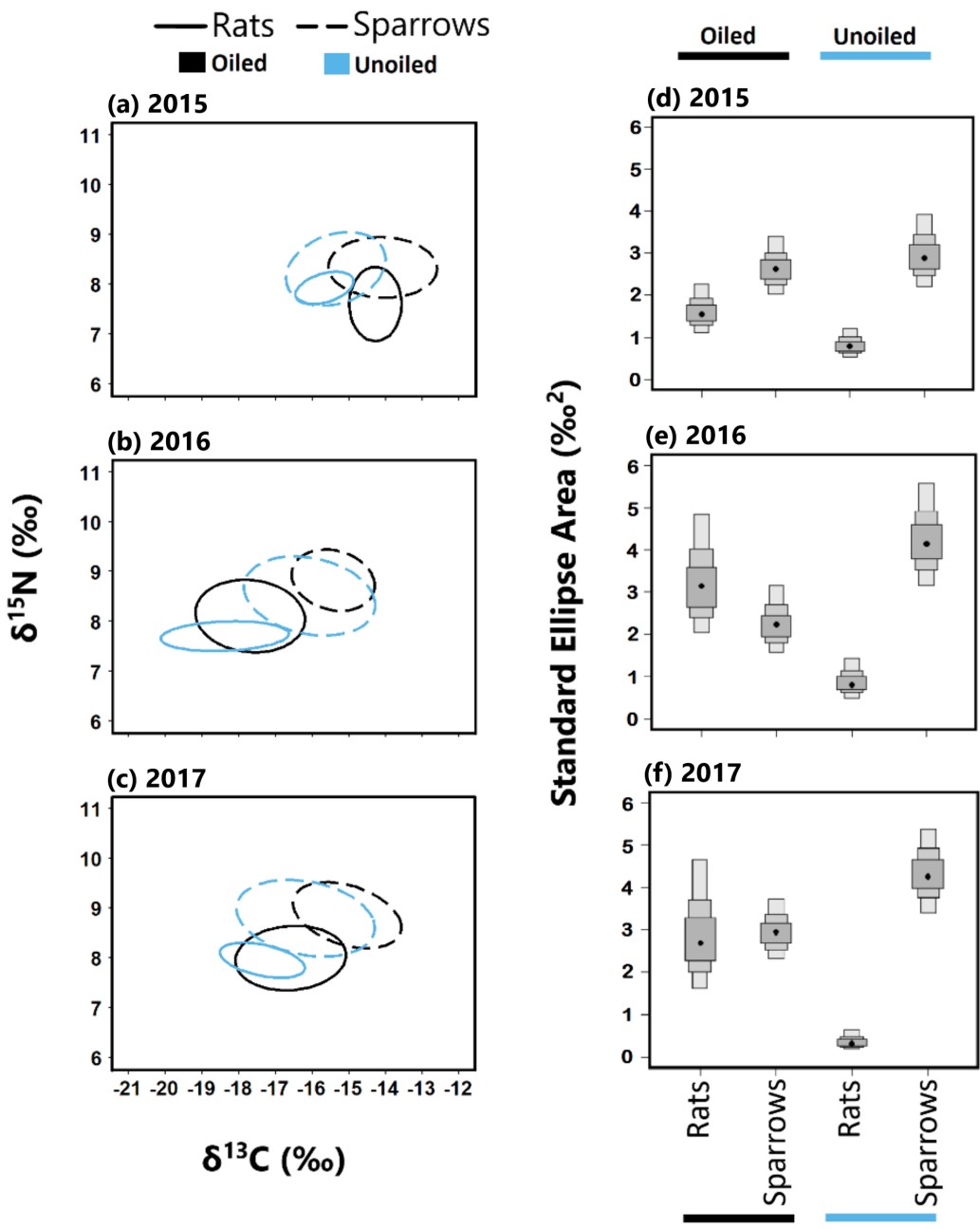

**Figure 4** **Stable isotope values and isotopic niche widths of seaside sparrow and marsh rice rat.** *Left panel*: Stable isotope values ($\delta^{13}C$ and $\delta^{15}N$) and isotopic niche widths of sympatric seaside sparrow (*A. maritima*) and marsh rice rat (*O. palustris*) blood samples collected annually, as indicated by standard ellipse areas (SEAc). *Right panel*: Bayesian derived estimates of standard elliptical area (SEA_b) for each species with associated 50, 75 and 95% credibility intervals.

**Table 2  Trophic niche overlap among seaside sparrow and marsh rice rat.** Total trophic niche overlap (posterior mean and 95% credible interval) among seaside sparrows (*A. maritima*) and marsh rice rats (*O. palustris*) collected between 2015 and 2017. Overlap is based upon ellipses encompassing 95% of the data and represents the percentage of the isotopic niche of species A within the isotopic niche of species B (*Swanson et al., 2015*). Two groups are considered to overlap asymmetrically when their lowest and highest estimates show no overlap.

| Species A | Oiling history | Species B | | | |
| | | *Rats* Oiled | *Rats* Unoiled | *Sparrows* Oiled | *Sparrows* Unoiled |
|---|---|---|---|---|---|
| **2015** | | | | | |
| *Rats* | Oiled | – | 38.1 (17–51) | 80.2 (71–93) | 83.0 (60–83) |
| *Rats* | Unoiled | 68.3 (49–75) | – | 96.1 (85–98) | 99.8 (99–100) |
| *Sparrows* | Oiled | 61.2 (49–70) | 36.0 (12–50) | – | 84.4 (79–91) |
| *Sparrows* | Unoiled | 51.2 (29–63) | 43.5 (28–68) | 83.7 (67–92) | – |
| **2016** | | | | | |
| *Rats* | Oiled | – | 43.4 (34–63) | 37.1 (25–50) | 85.9 (72–93) |
| *Rats* | Unoiled | 88.2 (76–97) | – | 18.6 (10–31) | 67.3 (50–83) |
| *Sparrows* | Oiled | 58.0 (47–90) | 9.9 (5–34) | – | 96.3 (96–100) |
| *Sparrows* | Unoiled | 73.5 (51–92) | 21.8 (11–44) | 69.9 (59–77) | – |
| **2017** | | | | | |
| *Rats* | Oiled | – | 46.1 (45–83) | 45.7 (34–66) | 81.2 (67–95) |
| *Rats* | Unoiled | 98.1 (88–99) | – | 22.1 (17–68) | 95.3 (90-99) |
| *Sparrows* | Oiled | 64.0 (39–80) | 5.8 (4–29) | – | 97.5 (90–99) |
| *Sparrows* | Unoiled | 63.9 (43–84) | 18.7 (11–40) | 72.6 (72–86) | – |

to freshwater ecosystems (*Hellmann, Wissel & Winkelmann, 2013*). The possible effects of environmental stressors on the trophodynamics of terrestrial vertebrates in coastal marshes are poorly understood (*sensu Olin et al., 2017*). This study presents insights into the foraging ecology, niche partitioning, and possible influence of oiling history on two ubiquitous vertebrates (the seaside sparrow and the marsh rice rat) in coastal saltmarshes of Louisiana. We observed a general pattern of interspecific differences in trophic position and aquatic vs. terrestrial resource use between sparrows and rats, likely due to their disparate morphologies and foraging behaviours. We found intraspecific differences in trophic niche width as well as variation in the degree and direction of trophic niche overlap and asymmetry between species with differing site oiling histories.

Trophic positions (hypothesis 1) differed between rats and sparrows in two out of three years examined (Fig. 3A). We found that the calculated modal trophic positions of sparrows (2.9 to 3.1; year =2015, 2016 and 2017) estimated in our study were lower than mean values calculated in a prior study (year = 2013 and 2014) in coastal Louisiana (3.5 $\pm$ 0.2 and 3.8 $\pm$0.2 across sites; *Olin et al., 2017*). The difference in trophic position between our study and the (*Olin et al., 2017*) study, could be attributed to differences in tissues analysed (i.e., blood versus liver), discrimination factors used ($\Delta^{13}$C = 0.39‰; $\Delta^{15}$N =3.4‰ in this study versus $\Delta^{13}$C = −0.5‰; $\Delta^{15}$N =3.0 ‰  used in the *Olin et al., 2017* study), and differences in the isotope values of aquatic ($\delta^{13}$C = −22.4‰; $\delta^{15}$N =7.2‰ in this study versus $\delta^{13}$C

$= -24.3‰$; $\delta^{15}N = 7.4‰$ in (*Olin et al., 2017*)) and terrestrial baselines ($\delta^{13}C = -12.2‰$; $\delta^{15}N = 4.1‰$ in this study versus $\delta^{13}C = -12.4‰$; $\delta^{15}N = 4.9‰$ in the (*Olin et al., 2017*) study). Few studies have sought to estimate the trophic position of rats. Even so, rats have been proposed to feed on similar food items and thus occupy a similar trophic position as sparrows (*sensu Post, 1981*). Our results suggest that rats occupy a slightly, but significantly, lower trophic position (2.5 to 2.8) relative to sparrows across the study region.

Overall, the largest temporal shifts in trophic position were observed in rats (whereby rats always occupied lower trophic positions than sparrows), suggesting that while rats are omnivorous, their omnivory changes over space and time. Temporal omnivory has been suggested as a key factor in maintaining ecological stability (*Kratina et al., 2012*). Temporal omnivory, whereby consumers incorporate prey from lower trophic levels into their diet when preferred prey become rare, slightly increases stability relative to the case of fixed omnivory (*Krivan & Diehl, 2005*). Such shifts in omnivory have been documented in freshwater (*Nakano & Murakami, 2001*) and desert vertebrates (*Soykan & Sabo, 2009*). Remaining questions include how the extent of this omnivory will change among individual sparrows and rats in future environmental perturbations.

Rats consumed a mixture of terrestrial and aquatic prey (Fig. 3), but generally consumed more aquatic-derived prey resources relative to sparrows (hypothesis 2; Fig. 3). This agrees with prior studies that suggest rats are omnivores that prefer aquatic organisms (*Sharp, 1967*) but are opportunistic feeders that shift their diet to utilize available resources (e.g., seeds and herbaceous plant parts, insects, and dicot vegetation; *Rose & McGurk, 2006*). The isotopic trophic niche of sparrows suggests they consume mainly terrestrial organisms, although they do access aquatic-derived prey resources as well. This is in agreement with the findings of other studies (*Olin et al., 2017*; *Johnson, Olin & Polito, 2019*), which indicate that sparrows in coastal Louisiana commonly forage on $C_4$ plant-consuming insects, as evidenced by stable isotope values typical of $C_4$ plants (Fig. 2; *Wainright et al., 2000*; *Wigand et al., 2007*). Our results agree with these prior studies and suggest that between 2015 and 2017 sparrows at our study area fed predominantly in a food-web based on plant and animal matter supported by *Spartina alterniflora* (*Post & Greenlaw, 2006*).

Sparrows exhibited wider trophic niches relative to rats at unoiled sites in all three years, and at oiled sites in 2015 only (hypothesis 3). This finding is compatible with the idea that sparrows, which glean food off both vegetation and the marsh sediment (*Post & Greenlaw, 2006*), usually exploit a broader range of prey resources relative to sympatric rats. This concept was confirmed by the partial support for our fourth hypothesis. Specifically, asymmetric trophic niche overlap was observed at unoiled sites in all three years. The trophic niche of rats overlapped to a much greater degree with those of sparrows (67.3 to 99.8%) at unoiled sites, than vice versa (18.7 to 43.5%). This suggests trophic niches are not fully segregated between these species despite observed differences in trophic position and resource use. In addition, given the wider trophic niches of sparrows relative to rats, there is a higher potential for rats to compete for food resources with sparrows, than *vice versa*. Even so, differences in trophic niche width as well as variation in the degree and direction of trophic niche overlap and asymmetry between species was less consistent at oiled sites. The difference in trophic niches could be ascribed to inter- and intraspecific

competition, which are known as major factors in determining the trophic niche width of coexisting species (*MacArthur, 1972*; *Kroetz, Drymon & Powers, 2017*; *Arribas, Touchon & Gomez-Mestre, 2018*). Density-dependent effects can also influence trophic overlap among coexisting species (*Van Beest et al., 2014*); however, there were no statistically significant differences in marsh rice rat density in 2015, 2016, and 2017 (Hart, unpublished data), nor between oiled and unoiled sites (Hart, unpublished data; A. Perez-Umphrey, unpublished data).

An additional goal of our study was to compare the trophic niches of sparrows and rats between and within species at sites with differing oiling histories to examine the ecological legacy of the 2010 DWH oil spill. Sample collection for our study (2015–2017) took place approximately five to seven years after the 2010 DWH oil spill. This lag and other factors (see below) makes identifying the proximate drivers of observed differences between oiled and non-oiled sites challenging. The redistribution of oil following Hurricane Isaac and the confounding effect of bay (Sansbois versus Batiste) i.e., effect of habitat differences between Bay Sansbois and Bay Batiste are also major problems that should be acknowledged. Regardless, we observe differences in trophic niche metrics between oiled and unoiled sites. For example, sparrows at oiled sites consumed more terrestrial prey resources (larger $\alpha$ values) than sparrows at unoiled sites in two out of the three sampling years. Similarly, rats at oiled sites consumed more terrestrial prey resources (larger $\alpha$ values) than rats at unoiled sites in 2015. In addition, rats had consistently larger trophic niche widths ($SEA_b$) at oiled sites relative to unoiled sites in all three years. According to optimal foraging theory, niche width will increase as the availability of specific resources decreases (*MacArthur & Pianka, 1966*; *Svanbäck & Bolnick, 2005*). Therefore, wider trophic niches of rats and the increased use of terrestrial prey resources at oiled sites could be explained by a decrease in aquatic prey resources preferred by rats. For example, using a meta-analysis approach, *Zengel et al. (2015)* found evidence that the DWH oil spill suppressed populations of aquatic organisms like fiddler crabs, a common diet item of rats, in oiled marshes with incomplete recovery as of 2014.

Isotopic niche widths of rats were higher at oiled sites compared to rats at unoiled sites (Fig. 4). The increase in niche width may show the subtle chronic effects of food web dynamics whereby the foraging behaviour of rats shifts to generalism (i.e., a niche expansion) at oiled sites, possibly in response to changes in prey availability/diversity. Researchers have reported persistent effects several years after the DWH oil spill whereby there were significant declines in diversity of marsh meiofauna and macrofauna (*Fleeger et al., 2015*; *Fleeger et al., 2019*), arthropods (*Bam et al., 2018*), fiddler crabs (*Zengel et al., 2015*) and snails (*Deis et al., 2020*). These changes in diversity and prey availability due to oil are not only limited to salt marshes but extend to other ecosystems. The Cosco Busan oil spill revealed that herring embryos that were directly exposed to oil and those at adjacent sites (but were unoiled) were significantly impacted by oil. Impacts on embryos at the sites were cardiac toxicity (sublethal) whereas impacts on embryos at adjacent sites was tissue death (*Incardona et al., 2012*). These data reveal an organism doesn't have to be present in an acutely oiled site to have negative impacts from the oil. Seven years after the DWH oil spill, impacts were observed for abundance, diversity (low diversity) and health

(low fecundity and larval output) of deep sea megafauna (*McClain, Nunnally & Benfield, 2019*). Elsewhere, four years after the DWH oil spill, oiled sites still exhibited depressed meiofaunal and macrofaunal diversity compared to unoiled sites (*Reuscher et al., 2017*). Further, monitoring of corals from 2010 to 2017 showed the majority of colonies exposed to oil had still not recovered by 2017 (*Girard & Fisher, 2018*) and it would take up to three decades for recovery from the DWH oil spill (*Girard, Shea & Fisher, 2018*). Taken together these studies suggest a lasting impact on faunal communities and a need to not only assess acute effects (as is common in studies of oil spills) but concomitantly study chronic indirect effects. Our results support the call by other researchers for long term monitoring of communities affected by DWH oiling effects in the Gulf of Mexico (*McClain, Nunnally & Benfield, 2019*).

Post-spill declines in the abundance of terrestrial invertebrates, such as insects and spiders, also were reported for Louisiana marshes after the DWH oil spill (*McCall & Pennings, 2012*; *Pennings, McCall & Hooper-Bui, 2014*; *Husseneder, Donaldson & Foil, 2016*). However, significant temporal, spatial, and taxon-specific variation in post-spill responses and recovery across studies and the potential for redistribution of DWH oil by hurricanes (*McCall & Pennings, 2012*; *Bam et al., 2018*) likely confound comparisons of the availability of terrestrial resources to sparrows and rats in our study. Specifically, Hurricane Isaac in 2012 is thought to have remobilized and redistributed oil that was deposited in inshore and offshore sediments such that some coastal locations that were initially oil-free became contaminated with petroleum hydrocarbons (*Turner et al., 2019*). In addition, oiling history is not the only factor that is likely to contribute to spatial and temporal differences in the trophic ecology of these two species. For instance, *Olin et al. (2017)* used stable isotope and fatty acid analysis to determine that marsh flooding due to Hurricane Isaac in 2012 had a stronger influence on the diets of sparrows than oiling status. Moreover, additional factors not examined here such as saltmarsh plant community composition, plant cover and biomass as well as elevation, inundation, and other hydrological aspects are likely to influence prey availability in combination with, or independent of, oiling histories (*Crosby et al., 2016*; *Olin et al., 2017*; *Boesch, 2020*).

## Conclusions and additional considerations

Our study identified a general pattern of interspecific trophic niche partitioning between rats and sparrows co-occurring within saltmarshes in coastal Louisiana. While we observed differences in trophic position and resources use, trophic niches were not fully segregated between rats and sparrows, with asymmetrical overlap between rats and sparrows. When comparing between sites with differing oiling histories we found both interspecific and intraspecific differences in the trophic niche width and the consumption of terrestrial vs. aquatic prey resources, although identifying the proximate causes of these differences is beyond the scope of this study. In addition, because trophic dynamics and environmental processes may be specific to individual sites or regions, the extension of our findings to other saltmarshes should be approached with some caution (*Nelson, Deegan & Garritt, 2015*). Even so, our findings provide reference information on respective trophic niche of

sparrows and rats that will aid in the evaluation of these species as bioindicators of oiling and other environmental stressors.

## ACKNOWLEDGEMENTS

We thank R. Strecker-Lau, W. Bam, K. Kjos, S. Wendt, M. Arias, M. Hart, S. T. Williams, J. Nierman, L. DiNunzio, S. Woltmann, A. Rietl, A. Chelskey, T. Hill, B. Kelly, C. Bourgeois, B. Kelly and S. Setta for field and laboratory assistance. The manuscript was approved by the Director of the Louisiana State University Agricultural Center as manuscript number 2021-241-35480.

### Funding

This research was supported by the Gulf of Mexico Research Initiative (GoMRI) to the Coastal Waters Consortium. This work was also supported by the National Institute of Food and Agriculture, U.S. Department of Agriculture, McIntire Stennis project LAB04066, LAB94169, and LAB94327. The funders had no role in study design, data collection and analysis, decision to publish, or preparation of the manuscript.

### Grant Disclosures

The following grant information was disclosed by the authors:
Gulf of Mexico Research Initiative (GoMRI) to the Coastal Waters Consortium.
National Institute of Food and Agriculture, U.S. Department of Agriculture, McIntire Stennis project: LAB04066, LAB94169, LAB94327.

### Competing Interests

The authors declare there are no competing interests.

### Author Contributions

- Sydney Moyo and Danielle Laguaite conceived and designed the experiments, performed the experiments, analyzed the data, prepared figures and/or tables, authored or reviewed drafts of the paper, and approved the final draft.
- Hayat Bennadji performed the experiments, analyzed the data, prepared figures and/or tables, authored or reviewed drafts of the paper, and approved the final draft.
- Anna A. Pérez-Umphrey and Allison M. Snider performed the experiments, authored or reviewed drafts of the paper, and approved the final draft.
- Andrea Bonisoli-Alquati, Jill A. Olin, Philip C Stouffer, Sabrina S. Taylor, Paola C. López-Duarte, Brian J. Roberts and Linda Hooper-Bui conceived and designed the experiments, performed the experiments, authored or reviewed drafts of the paper, and approved the final draft.
- Michael J. Polito conceived and designed the experiments, performed the experiments, analyzed the data, prepared figures and/or tables, authored or reviewed drafts of the paper, and approved the final draft.

## Animal Ethics

The following information was supplied relating to ethical approvals (i.e., approving body and any reference numbers):

All sampling protocols adhered to statutes of the Institutional Animal Care and Use Committee of the LSU AgCenter (IACUC permits: A2013-09, A2015-04, A2016-06).

## Field Study Permissions

The following information was supplied relating to field study approvals (i.e., approving body and any reference numbers):

The capture and handling of all animals was approved by the United States Fish and Wildlife Service (collecting permits MB095918-0 and MB095918-1; USFWS Federal Bird Banding permit 22648; and, Louisiana Department of Wildlife and Fisheries scientific collecting permits LNHP-15-039, LNHP-16-056, LNHP-17-064, LNHP-15-033, LNHP-16-048, LNHP-17-039).

## Data Availability

Data are publicly available through the Gulf of Mexico Research Initiative Information & Data Cooperative (GRIIDC) at: Polito, Michael, J., Moyo, Sydney, Olin, Jill, Johnson, Jessica, Lopez-Duarte, Paola, Roberts, Brian, Hooper-Bui, Linda, Taylor, Sabrina, Stouffer, Philip. 2019. Carbon and nitrogen stable isotope values of marsh rice rats, seaside sparrows, primary consumers and basal carbon sources from southern Louisiana salt marshes, 2015-05-01 to 2017-06-30. Distributed by: Gulf of Mexico Research Initiative Information and Data Cooperative (GRIIDC), Harte Research Institute, Texas A&M University–Corpus Christi. DOI: 10.7266/n7-6277-1216.

https://data.gulfresearchinitiative.org/data/R6.x808.000:0031.

## Supplemental Information

Supplemental information for this article can be found online at http://dx.doi.org/10.7717/peerj.11392#supplemental-information.

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
