# Peer review of "Stable isotope analyses identify trophic niche partitioning between sympatric terrestrial vertebrates in coastal saltmarshes with differing oiling histories"

_PeerJ, doi:10.7717/peerj.11392_

## Round 0.1 · original submission · Major Revisions

The paper is within the scope of the journal and will make a solid contribution after major changes. Reviewers suggest changes that will improve the quality if the paper. Please follow carefully their suggestions and resubmit the paper.

Reviewer 1 ·

Basic reporting

The writing is clear with only a few minor grammatical mistakes. The literature is cited appropriately to provide supporting information or support inferences. Results are self-contained and relevant to hypotheses, but I suggest a reworking of hypotheses or objectives which will affect the organization of subsequent sections (see general comments to authors). Figures and tables are appropriate and professionally made. However, I suggest moving Fig S1 to the main body of the paper following the map figure, thus making it Fig 2 and shifting the remaining figures one spot. In that figure, the primary producer or consumer data should either be overlain on the bird or rat panels (preferred) or come as panel a. If included as a separate panel, the y-axis scale should be the same as the other other two taxon-specific panels.

Experimental design

This research is original and within the aims and scope of the journal. The research questions are well defined, although I think the order and focus should be shifted (see general comments to author). Stable isotope and statistical methods are described in sufficient detail.Appropriate inferences are drawn from the data, but I feel the authors have been a little too conservative in discussing their findings.

Validity of the findings

All data are presented in either raw or summarized forms. The design is robust and appropriate controls exists within the authors' field experiment. Conclusions are mostly well-stated, but data need to be interpreted more completely in the Discussion. The authors tend to restate hypotheses and results in the Discussion and only draw limited and conservative inferences from their work.

Additional comments

The Moyo et al. study is well-designed with a sound approach. It fits well within the scope of PeerJ and is likely to be well-cited once published. The paper is generally well-written, with only a few minor grammatical or punctuation errors, and the authors do not attempt to draw inferences beyond their data. I do think some restructuring of the manuscript prior to publication will improve its readability and make it more impactful. Part of that involves the tail end of the introduction and the remaining reorganization suggestions are for the discussion. I detail these more substantive suggested changes below, followed by a list of more minor suggested edits.

The introduction focuses on the usage of indicator species to examine anthropogenic stressors and then moves into the specific utilization of seaside sparrows and rice rats as indicator species of Deepwater Horizon Oil Spill (DWH) impacts on Louisiana (LA) marsh food webs. However, the authors actually do not explicitly state that is how they are employing their two study species; they leave out a much needed statement about food webs. I feel they should also state that in the case of many anthropogenic stressors researchers often focus on direct acute affects versus more indirect chronic effects. Their study is an excellent example of the latter. If they stated that more explicitly I think it would allow them to more easily draw inference from their data in the discussion.

Another area of the introduction that I feel needs some attention is the statement of hypotheses. As written, they are a mix of null hypotheses and predictions. To avoid that confusion, why not just state the objectives of the study? Toward that end, I would start with the first objective being to estimate d13C and d15N for basal endmembers and primary consumers. The second objective would be to estimate species-specific trophic position and isotopic niche width for seaside sparrows and rice rats from unoiled (control) sites. The final objective would be to test for differences in basal endmembers as well as between seaside sparrows and rice rats, between control and oiled sites. I would focus less on the idea that these two species likely interact trophically, but instead focus on their roles as common, generalist mesopredators in the LA marsh ecosystem which makes them ideal model species to examine indirect chronic effects of the DWH on LA marsh food webs. And by stating the approach via objectives, it removes the need or tendency to repeat the hypotheses in the discussion, or list them parenthetically in the results.

One reason to suggest the first objective was to examine differences in basal endmembers and primary consumers between terrestrial and aquatic habitats, as well as between control and oiled sites (which are the preferred terms versus oiled and unoiled), is because those data are what the entire study hinges upon. The authors should be commended for conducting that sampling as too often authors attempt to borrow literature values instead of conducting contemporary sampling. It is actually unclear whether basal endmembers and primary consumers were sampled at each site. If both were sampled as suspected, then and the data for controlled and oiled sites are combined in the biplot in Fig S1C. I suggest this figure be moved to the main body of the manuscript and appear as Fig 2, just after the map figure. If they exist, the data for control versus oiled sites should be displayed separately, either as Panel A of that figure or overlain on the sparrow or rat panels. One reason to display control and oiled sites separately is that if petrocarbon from DWH was assimilated into the marsh food web at oiled sites, then the d13C signature should be depleted. Regardless, differences in d13C and d15N should be tested with ANOVA between control and oiled sites to determine whether significant differences existed in basal endmembers or primary consumers.

Different terminology is utilized throughout the intro, methods, and discussion to refer to trophic niche or isotopic niche. Mostly, the term “foraging niche” is utilized, but this is a misnomer. Elsewhere, “trophic niche” or “foraging trophic niche” are utilized, but the preferred term is “trophic niche.” When the SEA or isotope data are being utilized to draw inference about the trophic niches of the two model species, then “isotopic niche” should be utilized, which should be stated is a proxy for trophic niche based on d13C and d15N data.

One bit of information that is not included in the results is the size range of sparrows and rats. Did the authors test for effects of animal size on d13C or d15N values? Ontogenetic shifts in diet often can add a source of error that should be accounted for. This can be accomplished by detrending the data or adding animal size as a covariate in statistical analyses.

In the discussion, there is too much restatement of hypotheses from the intro and too much restatement of results. Instead, the authors should focus on the inferences from the significant findings of their study. They demonstrated significant differences in trophic position, isotopic niche breath, and even trophic overlap between species between control and oiled sites. Instead of focusing on the caveats associated with those findings, I would focus on the remarkability of persistent food web effects despite the time since the spill, weathering of oil, and other factors that add considerable process error to the system. Furthermore, there are interesting patterns in there data that suggest shifts to more generalism even 5-7 years after the DWH in their study species. While there is touched upon in the discussion, very little ecological theory is discussed as to why this may be an important indication of food web effects. The authors state that their sparrows and rats are being utilized as indicator species, but they do not do much in the form of stating what they are actually indicating.

Other authors have reported persistent food web effects of the DWH in different habitats or systems than the marsh, but the Moyo et al. do not cite or discuss that literature, which hurts the discussion. Lastly, I would return to the theme of acute direct versus chronic indirect of the spill. Their study nicely demonstrates the potential for the latter in the LA marsh food web, but they fail to mention or discuss that.

Minor comments:
line 32: “species-specific” is redundant with niche

lines 35-38: This is written as if it was the main objective of the study, but the paper does not support that as written.

line 43: “though” should be replaced with “although” throughout.

line 54: Should be Deepwater Horizon Oil Spill (DWH) here, then DWH thereafter, with Deepwater Horizon italicized.

lines 55-56: “which lasted for several months” appears twice

line 61: What is meant by “penetrated deep into the marsh?” What dimension is being referred to here and what is meant by deep?

lines 62-65: Subordinate clause (about oiling) does not match the main clause here (about general contaminant effects).

line 68: Are these animals truly ubiquitous, or just common in these habitats?

line 69: For example indicates this refers to the previous statement, but here that is not true.

line 80: I am not sure what a life history ecology is. Do you mean feeding ecology and life history?

line 81: This is the first time you have used trophic versus foraging niches. Trophic is more accurate and should be utilized throughout.

line 83: What about morphology and physiology? There are intrinsic factors that are overlooked here.

line 115: See comment about line 83 text. Here other factors are mentioned.

line 118: Should be hypothesized, past tense, but I think it would be better to phrase these as study objectives instead of hypotheses.

line 137: (Cooney et al. 2015)

line 142-143: …were sampled on private or public land…

line 152: …2015, 2016, and 2017…

line 152: The saltmarsh in this area is either dominated by smooth cordgrass, Spartina alterniflora, or co-dominated by smooth cordgrass and black needlerush, Juncus roemerianus.

line 162: Sites not Site

line 167: Need to indicate mean (+/- dispersion measure) here for PAH values.

line 233: represented BY the…

line 233: encompassing approximately 40%...

line 251: Need to present basal endmember and primary consumer data first.

line 252: Trophic position not positions

line 269: Why add the word “previously” here?

line 269: It is unclear what tables S1-S3 are actually showing because they lack captions.

line 275: isotopic niche instead of isotopic foraging niche

lines 276-280: Strike second and fourth sentences of this paragraph.

line 291: No overlap measure provided in the table for this first set of CIs.

lines 296-297: … the feeding ecology of terrestrial omnivores IS often flexible and dependent…

line 312: Not consistent; half a trophic position lower. Why? Were the same tissues analyzed in both studies? Where same TDFs used in both studies? Did basal endmember values differ?

lines 318-320: Run-on sentence

line 334: glean food?

lines 340-341: This suggests trophic niches are not fully segregated between these species despite observed differences in trophic position and resource use (alpha).

lines 342 and 343: marsh not march

line 365-366. Post-spill declines in the abundance of terrestrial invertebrates, such as insects and spiders, also were reported for Louisiana marshes after DWH.

lines 383-385: See comment above for lines 340-341.

·

Basic reporting

The writing in this paper meets these criteria. I noted only a couple of little problems, below. Literature is cited correctly and the paper directly addresses specific hypotheses.

Experimental design

Research is well designed and appropriately sampled. Statistical tests are appropriate.

Validity of the findings

Experimental data are well analyzed and significance of the results is well tested, documented and explained.

Additional comments

Clear hypotheses are articulated and I like the way the results and discussions sections are organized to address each one.

Lines 296-297-- In coastal marsh ecosystems, the feeding ecology of terrestrial omnivores are often flexible and are dependent on environmental conditions (Ruiter, Wolters & Moore, 2005). Relative to what other environments?

Discusion line 354-357. Interesting that the consumption of terrestrial resouces in oils sites was due to less abuncance of aqutic prey items even 5 years after the oil spill. Is this in the abstract? Seem important.

Repeated phrase lines 54-57. Specifically, the Deepwater Horizon disaster, which lasted for several months, was the largest (> 700 000 m3 of crude oil) marine oil
spill in the history of the United States of America, which lasted for several months (Crone &
57 Tolstoy, 2010).

·

Basic reporting

I found parts of the manusript confusingly worded and at times had a hard time sorting out the authors’ intention. I also think that the manuscript would be greatly improved by adding some basic background on the sites and the two species targeted.

Experimental design

The authors provided insufficient detail on the study sites and how similar or different the individual sites were, both within and between years. A critical step was to combine all of the oiled and all of the unoiled sites, but no justification was provided other than the need to increase sample size. At the least, some assessment of differences among sites should be provided.

Validity of the findings

I had a hard time seeing some of the patterns described. That’s not to say that the findings are incorrect, but at the least the figures and text need to be reworked to provide a clear and consistent scientific message.

Additional comments

This is an interesting set of isotopic measurements intended to help shed light on the impact of oil deposition on marsh foodwebs. Unfortunately, I found many parts of the manuscript and its figures difficult to follow and was frustrated by having to parse the text and figures carefully to try to figure out the authors’ message. This manuscript does have promise, but I recommend a thorough revision of the text and figures, and addition of some critical contextual information on the populations and communities studied.

A number of more specific comments and suggestions follow.

Abstract, lines 42-43: This sentence is an example of the sort of imprecision I struggled with. Should this sentence be read to mean that sparrows from unoiled sites had wider niches than rats collected from unoiled sites (ignoring the animals from oiled sites) or that sparrows from all sites had wider niches than rats from the unoiled sites. As written, the sentence can be read either way, leaving the reader to puzzle out the meaning. The next sentence does imply that the first reading is the correct one, but many readers will come to that only after spending time and energy trying to sort out the meaning of the earlier sentence.

Lines 67-69: A few words about the life history of the sparrows and rats would help. Do they have similar life spans? Do they mature at similar rates?

Lines 75-76: Did oiling alter sparrow and rat abundance along with the abundances of their potential prey? If so, how do you distinguish between impacts of intra- and inter-specific competition?

Line 80: Indeed! A bit more background on the life histories of the two species would be very helpful here.

Lines 124-125: I found this sentence very confusing! The absolute overlap of species A’s niche with that of species B has to be equivalent to the overlap of species B’s niche with that of species A. What actually matters, and what the authors are after here, is the degree of niche overlap relative to a species’ overall niche. That is, it’s the relative, not absolute overlap that matters, but that subtlety is missing here.

Lines 130-140: A couple of sentences on the life histories and populations of the two species would be helpful here. What is their typical life span? Were the animals sampled in 2017 descendants of the ones sampled in 2015, or part of the same cohort? All of these details are important context for discussing the trophic ecology of these animals.

Lines 164-165: I can’t sort out the dimensions of the plot given this description. The bit in parentheses is particularly cryptic.

Lines 169-171: What about the animals? Are they long-lived enough that the animals sampled for this study may have also been present in 2010?

Lines 173-174: Many more details are needed here. For example, were collections carried out at the same time of year in all years? Were the animals collected all of similar size? Were the size/mass of the animals measured? Were the animals sacrificed after sampling or released?

Line 198: The equation is missing a factor of 1000

Lines 210-213: Did you first test for differences among sites? I realize that sample numbers may have been low in some cases, but this should be done before lumping data in this way.

Lines 244-247: This definition makes sense and should have been presented earlier to avoid the confusion I noted above (lines 124-125).

Line 262: I don’t see this at all. If anything, 2015 looks like the outlier, with what look like appreeciably higher alpha values than in 2016 and 2017. In any case, it’s hard to make this comparison visually given the layout and the visual confusion created by the asterisks, which have nothing to do with this comparison. See below for other comments on Figure 2.

Lines 285-287: This should be evident in Fig 3, but only 2015 shows the asymmetric overlap described in the text. In fact, there’s minimal if any overlap between the species’ isotopically defined niches in 2016 and 2017. This deserves a bit more description and explanation.

Lines 287-293: It’s interesting that there appears to be no niche overlap at all in 2016; this seems interesting and relevant and should be noted.

Lines 309-310: This statement is a bit misleading since the two spp. differed significantly between oiled sites in 2015. In other words, only one of the six comparisons (2015, unoiled sites) showed no significant difference in trophic position.

Line 320: Fig. 2, not Fig. 3 is the relevant one here.

Lines 338-340: As noted above, this isn’t evident in Fig. 3, where two of the three years show minimal if any niche overlap (2016 and 2017) at unoiled sites.

Lines 342-346: What about the role or intraspecific competition in this system? Did the population density of either species vary between years? For that matter, did food abundance vary between years?

Lines 377-380: These are factors that really should have been addressed earlier! I was curious about differences among sites when I saw that they had been grouped simply by oiling history (see above re: lines 210-213), and I’m now left to wonder how similar the different sites actually were. At the least, some assessment of the community, and the population densities of the two target species and their likely food sources would be helpful.

Figure 2: Why the modal value? There may well be a good reason for this, but wouldn’t it make more sense to show the mean or median value? I’m also puzzled by the choice of modal value for a continuous variable; were the data binned in some way?

Figure 2: The asterisks mark cases where the two species differ significantly within a habitat, but the figure is also used to discuss differences between years within a habitat (e.g., Line 262). This makes for a confusing figure that forces the reader to parse the data for him/herself. I strongly recommend exploring clearer ways to present the data graphically.

Figure 2: The sparrow and rat symbols should be offset (jiggled) slightly so that the error bars can be clearly distinguished. As presented, it’s hard to tell which bars go with which symbols.

Figure 3: The numeric labels are far too small for legibility, and the d13C label on the horizontal axis really should be closer to the horizontal axis of panel C.

============End of Review====================

---

## Round 0.2 · accepted · Accept

I consider that the authors have made the changes and include suggestions made by the reviewers, so the paper is ready to be accepted for publication.

·

Basic reporting

The authors have revised the paper following reviewer guidance and in my opinion the paper is acceptable for publication.

Experimental design

The authors have revised the paper following reviewer guidance and in my opinion the paper is acceptable for publication.

Validity of the findings

The authors have revised the paper following reviewer guidance and in my opinion the paper is acceptable for publication.

Additional comments

The authors have revised the paper following reviewer guidance and in my opinion the paper is acceptable for publication.

·

Basic reporting

This revised version is well written and much clearer than the original submission.

Experimental design

The authors have added detail on the study sites and have successfully reworked the text for clarity.

Validity of the findings

The revised figures and manuscript organization address my earlier concerns with the original submission.

Additional comments

This is an interesting set of isotopic measurements intended to help shed light on the impact of oil deposition on marsh foodwebs. This revision addresses the concerns I had with the original submission. I have only minor comments and suggestions for the authors:

Lines 225-227: how was the POM removed from the filters? Since particles are trapped in the filter matrix as well as on the filter surface, it’s common to simply combust the filter rather than trying to remove material from it.

Lines 372-381: Did the absolute isotope values differ between studies, or was this contrast arise strictly in the calculation of TP?

Figure 2: It would be easier to compare between oiled and unoiled sites visually if unfilled versions the same symbols (square, circle, triangle) were used (colored blue, of course).

============End of Review====================